# NIR-II emissive anionic copper nanoclusters with intrinsic photoredox activity in single-electron transfer

Li-Juan Liu[1,2,6], Mao-Mao Zhang[1,6], Ziqi Deng[1], Liang-Liang Yan[1,3,4], Yang Lin[1], David Lee Phillips [1], Vivian Wing-Wah Yam [1,3,4] & Jian He [1,3,5] ✉

Ultrasmall copper nanoclusters have recently emerged as promising photocatalysts for organic synthesis, owing to their exceptional light absorption ability and large surface areas for efficient interactions with substrates. Despite significant advances in cluster-based visible-light photocatalysis, the types of organic transformations that copper nanoclusters can catalyze remain limited to date. Herein, we report a structurally well-defined anionic $Cu_{40}$ nanocluster that emits in the second near-infrared region (NIR-II, 1000–1700 nm) after photoexcitation and can conduct single-electron transfer with fluoroalkyl iodides without the need for external ligand activation. This photoredox-active copper nanocluster efficiently catalyzes the three-component radical couplings of alkenes, fluoroalkyl iodides, and trimethylsilyl cyanide under blue-LED irradiation at room temperature. A variety of fluorine-containing electrophiles and a cyanide nucleophile can be added onto an array of alkenes, including styrenes and aliphatic olefins. Our current work demonstrates the viability of using readily accessible metal nanoclusters to establish photocatalytic systems with a high degree of practicality and reaction complexity.

Over the past few decades, the use of atomically precise coinage-metal nanoclusters in catalytic applications has grown rapidly due to their discrete energy levels, unique electronic configurations, and structure-dependent activities[1–5]. With energy quantization manifested in a low energy gap[6], nanosized metal clusters typically exhibit a light absorption ability spanning the ultraviolet to visible or even near-infrared ranges[7–9], making them potentially effective photosensitizers for the development of photoinduced organic transformations[10,11]. Despite significant progress in cluster-based photocatalysis involving reactive oxygen species[12–15], other types of photoinduced coupling reactions, particularly those requiring transition metals to facilitate bond-forming elementary steps, remain difficult to achieve[16]. It is envisioned that copper nanoclusters may become suitable catalyst candidates for advancing this research direction[11,17–19], mainly because

copper is one of the earth-abundant metals and is capable of promoting various carbon–carbon and carbon–heteroatom bond formation via radical recombination pathways[20–25].

In recent years, diverse photoactive copper(I) complexes have been exploited to enable radical cross-coupling reactions by modulating either X-type ligands originating from external nucleophiles[21,26,27] or L-type nitrogen/phosphorous-donor ligands (Fig. 1a)[23,28,29]. In order to improve the synthetic utility of visible-light photocatalysis, the latter strategy, which permits the incorporation of a variety of nucleophiles in the coupling processes, is increasingly employed[30,31]. Regarding the design of cluster-based photocatalysts, external carbazolide (X-type) ligands have been utilized to generate photoactive copper(I) species that facilitate the Ullmann C–N coupling of aryl halides (Fig. 1b)[32]. However, without the prior coordination of

[1]Department of Chemistry, The University of Hong Kong, Hong Kong, China. [2]Chemistry and Chemical Engineering of Guangdong Laboratory, Shantou, China. [3]State Key Laboratory of Synthetic Chemistry, The University of Hong Kong, Hong Kong, China. [4]Institute of Molecular Functional Materials, The University of Hong Kong, Hong Kong, China. [5]Materials Innovation Institute for Life Sciences and Energy (MILES), HKU-SIRI, Shenzhen, China. [6]These authors contributed equally: Li-Juan Liu, Mao-Mao Zhang. ✉e-mail: jianhe@hku.hk

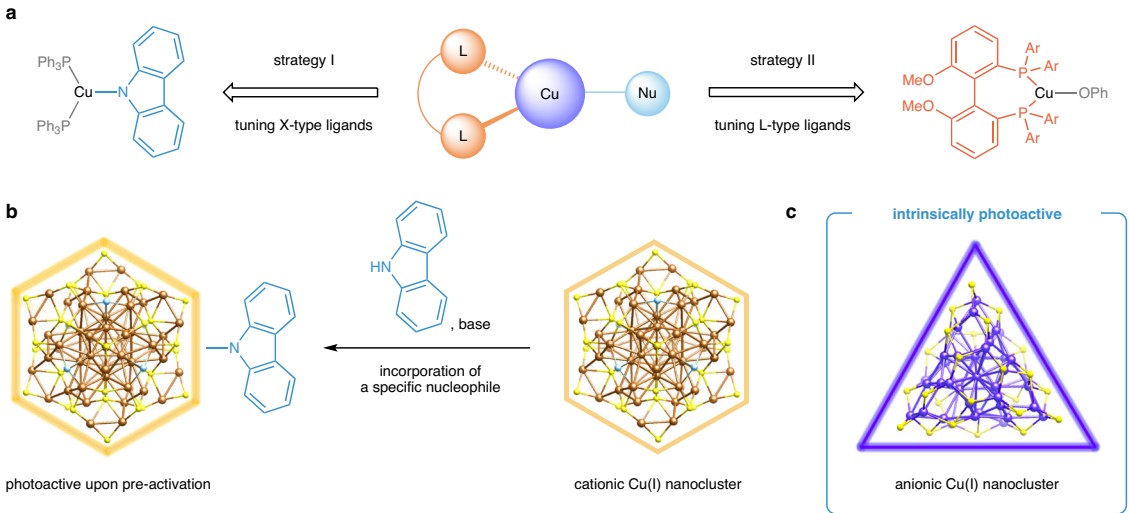

**Fig. 1 | Design of copper-based photoredox catalysts for radical couplings.**
**a** Constructing photoactive copper(I) complexes through the modification of X- or L-type ligands. **b** Using carbazolide nucleophiles to convert non-emissive cationic copper nanoclusters into photoredox-active species. **c** Preparation of anionic copper nanoclusters with intrinsic photoredox activity.

specific nucleophiles, the majority of emissive copper nanoclusters, which are either cationic[33–35] or neutral[36–38], cannot initiate single-electron transfer with electrophiles to produce organic radicals under visible-light irradiation conditions. Due to their potentially enhanced reducing ability in photoexcited states, we anticipated that emissive anionic copper nanoclusters could serve as suitable photoredox catalysts for developing visible-light-induced radical coupling reactions with broad substrate scope without the need for pre-activation with external nucleophiles.

In this study, 2,4-dimethylbenzenethiolate (2,4-DMBT) is employed as a protecting ligand to synthesize a highly scalable anionic $Cu_{40}$ nanocluster, $[Cu_{40}H_{17}(2,4-DMBT)_{24}](PPh_4)$ (denoted as $Cu_{40}$-H NC), with excellent air and moisture stability (Fig. 1c). Structural analysis suggests a helical arrangement of tessellated polyhedral units surrounding a tetrahedral $Cu_4$ core, which confers a $C_3$ axis on the new nanocluster. Importantly, intrinsically photoactive $Cu_{40}$-H NC with strong light absorption from UV to visible regions can efficiently promote electron-transfer-mediated three-component cyanofluoroalkylation reactions of alkenes, fluoroalkyl iodides, and trimethylsilyl cyanide under blue-LED (456 nm) irradiation. The nanocluster-based photocatalytic process is compatible with a wide range of alkenes and fluoroalkyl iodides and tolerates a variety of functional groups.

## Results and discussion
### Synthesis and characterization of photoactive copper nanoclusters
The air- and moisture-stable $Cu_{40}$ nanocluster was synthesized by directly reducing $[Cu(CH_3CN)_4]PF_6$ with $NaBH_4$ in the presence of 2,4-dimethylbenzenethiol (2,4-DMBTH) and tetraphenylphosphonium bromide (PPh_4Br) under ambient conditions. Other less sterically hindered benzenethiols and aliphatic thiols, such as 4-methylbenzenethiol and 2-phenylethanethiol, failed to produce $Cu_{40}$ nanoclusters, highlighting the significance of weak interactions from 2,4-DMBT in the nanocluster formation. After one week of slow vapor diffusion of hexanes into a dichloromethane-toluene solution, dark red crystals of $Cu_{40}$-H NC suitable for single-crystal X-ray diffraction (SCXRD) measurements were obtained on a gram scale in a high yield of 60% (based on Cu) (Supplementary Figs. 1 and 2). Elemental mapping shows that Cu, S, and P elements are uniformly distributed throughout its block-like crystals (Supplementary Fig. 3).

According to the SCXRD analysis, $Cu_{40}$-H NC crystallizes in the monoclinic $P2_1/c$ space group with a pair of enantiomers, each of which

contains 40 copper atoms, 24 2,4-DMBT ligands, and one tetraphenylphosphonium cation (Fig. 2a and Supplementary Fig. 4). The total structure of $Cu_{40}$-H NC racemates and their packing mode in the same crystal are depicted in Supplementary Figs. 5 and 6. The $Cu_{40}$ kernel and its protective thiolate shell share the same $C_3$ axis which passes through the central copper atom (Fig. 2b). As illustrated in Fig. 2c, the kernel structure can be divided into three primary components: a $Cu_{28}$ motif resembling a double three-bladed propeller (right), a $Cu_9$ hexagonal pedestal (left), and a belt-like $Cu_3$ unit (middle). The $C_3$-symmetrical $Cu_{28}$ motif is viewed as a tessellated polyhedron with a tetrahedral $Cu_4$ core at its center (Fig. 2d). At the top of the copper nanocluster, there are three rhombic pyramid-shaped $Cu_5$ units sharing the same vertex atom with the $Cu_4$ tetrahedron (Fig. 2e). The remaining three vertices at the bottom of the $Cu_4$ tetrahedron connect three twisted rhombohedral $Cu_6$ units in the same layer, each of which shares another vertex with one of the rhombic pyramids (Fig. 2f). The average Cu−Cu distances in the tetrahedral, rhombic pyramidal, and rhombohedral building units are 2.76, 2.88, and 2.67 Å, respectively[35,39]. The pedestal-shaped $Cu_9$ unit with an average Cu−Cu distance of 2.74 Å produces an equilateral triangle at its central position; each vertex of the triangle is capped by two additional copper atoms (Fig. 2g and Supplementary Fig. 7). The three copper atoms at the waist are positioned above the square faces of the trigonal prism formed by connecting the $Cu_{28}$ and $Cu_9$ units (Fig. 2h, i).

The surface thiolate ligands can be classified into four groups based on their distribution in the protective shell. Three identical ligand groups composed of seven thiolates stabilize the kernel structure by connecting the three primary components as well as the building units within the $Cu_{28}$ motif. The aryl substituents of each group are assembled via π−π stacking and C−H···π interactions (Supplementary Fig. 8). The fourth group, which consists of the remaining three thiolates in the shell, shields the copper atoms on the hexagonal pedestal. The characteristic C−H···π interactions[40] are identified in the trimeric ligand assembly (Supplementary Fig. 9).

Due to the low electron density surrounding hydrogen atoms, it is difficult to determine their exact number and positions in the crystal lattice through SCXRD. Therefore, electrospray ionization time-of-flight (ESI-TOF) mass spectrometry, nuclear magnetic resonance (NMR) spectrometry, and density functional theory (DFT) analysis were employed to further characterize the hydrides in the copper nanocluster. As shown in Fig. 3a, $Cu_{40}$-H NC features a monoanionic peak at $m/z = 5851.28$ Da (cal. 5851.31 Da) in the ESI-TOF mass

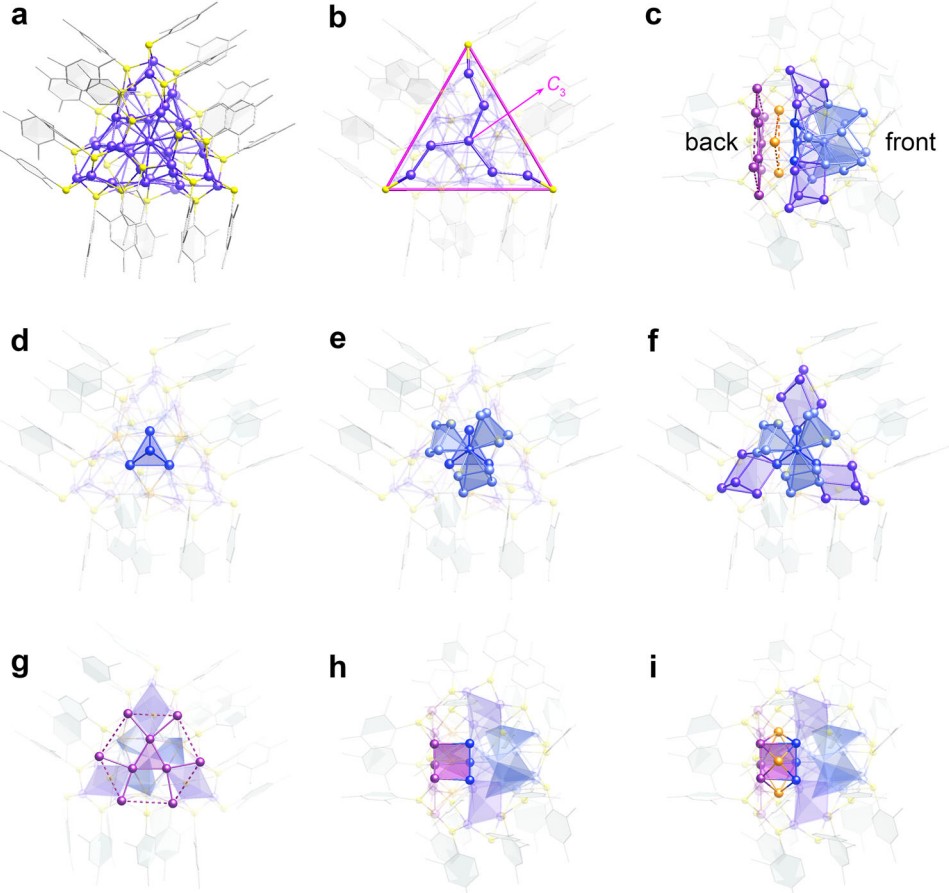

**Fig. 2 | X-ray crystallographic structure anatomy and symmetry analysis.**
**a** Total structure of Cu$_{40}$-H NC. **b** Cu$_{40}$ kernel and ligand arrangements with a $C_3$ axis. **c** Structure anatomy of the kernel with a side view. **d** Tetrahedral Cu$_4$ unit of the Cu$_{28}$ motif viewed from the front. **e** Adding three rhombic pyramids to the central vertex of the Cu$_4$ tetrahedron. **f** Adding three rhombohedra based on three edges formed by the vertices of the tetrahedron and rhombic pyramids. **g** Cu$_9$ hexagonal pedestal viewed from the back. **h** Connecting the central atoms of the hexagonal pedestal to the three vertices of the tetrahedron to generate a Cu$_6$ trigonal prism. **i** Capping the square faces of the trigonal prism with three copper atoms at the waist. Color labels: Cu, violet, magenta, blue, light blue, and orange; S, yellow; C, gray. All hydrogen atoms and the phosphonium cation are omitted for clarity.

spectrum, confirming the presence of [Cu$_{40}$H$_{17}$(2,4-DMBT)$_{24}$]$^-$ anion. The counterion (PPh$_4^+$) was also successfully identified by ESI-TOF mass spectrometry in a positive mode and $^{31}$P NMR studies (Supplementary Figs. 10 and 11). The deuteride analogue [Cu$_{40}$D$_{17}$(2,4-DMBT)$_{24}$](PPh$_4$) (denoted as Cu$_{40}$-D NC) was prepared using the same procedure as Cu$_{40}$-H NC, with the exception that NaBD$_4$ was utilized in place of NaBH$_4$. As predicted, the ESI-TOF mass spectrum of Cu$_{40}$-D NC displays a major signal at $m/z$ = 5868.46 Da (cal. 5868.45 Da), which corresponds to the [Cu$_{40}$D$_{17}$(2,4-DMBT)$_{24}$]$^-$ anion (Fig. 3b). The $m/z$ difference between [Cu$_{40}$H$_{17}$(2,4-DMBT)$_{24}$]$^-$ and [Cu$_{40}$D$_{17}$(2,4-DMBT)$_{24}$]$^-$ (5868.46 − 5851.28 = 17.18) reveals that the Cu$_{40}$ nanocluster contains 17 hydrides[41]. Moreover, five distinct $^2$H NMR signals ranging from 5.05 to −1.67 ppm in the spectrum of Cu$_{40}$-D NC but not in that of Cu$_{40}$-H NC support the existence of hydride species in the cluster (Supplementary Fig. 12).

To precisely determine the geometric positions of the 17 hydrides in Cu$_{40}$-H NC, DFT calculations were performed on the basis of the symmetry of the entire kernel as well as the four types of copper hydride species identified by SCXRD (Supplementary Fig. 4)[42,43]. Through the optimizations involving the simplification of 2,4-DMBT ligands to SCH$_3$ groups, 17 hydrides with seven distinct coordination environments were successfully assigned to the [Cu$_{40}$H$_{17}$(SCH$_3$)$_{24}$]$^-$ anion (Fig. 3c), producing a single $C_3$ axis in their symmetrical distributions. The capping hydrides (μ$_3$-H) and interstitial hydrides (μ$_4$-H and μ$_5$-H) are clearly discernible in Fig. 3d. The seven μ$_3$-H ligands bind

to seven Cu$_3$ triangles with Cu−H distances in the range of 1.64−1.80 Å[44]. One triangle remains at the center of the hexagonal pedestal; three stay in the layer between the Cu$_9$ pedestal and the Cu$_{28}$ polyhedral motif; three connect the Cu$_5$ rhombic pyramids and the Cu$_6$ rhombohedra within the Cu$_{28}$ motif (Fig. 3e). As illustrated in Fig. 3f, there is one μ$_4$-H ligand in the central Cu$_4$ tetrahedron and three in the grooves between the base of the Cu$_{28}$ motif and the belt-like Cu$_3$ unit. In the three rhombohedra, three additional μ$_4$-type interstitial hydrides are located in close proximity to the faces shared with the central tetrahedron (Fig. 3g). The Cu−H bond lengths in the μ$_4$-coordination mode range from 1.66 to 1.91 Å. Lastly, the three μ$_5$-H ligands are found in the cavity of three square pyramids which are constructed by the kernel atoms in the Cu$_{28}$ motif (Fig. 3h), with an average Cu−H bond length of 1.88 Å.

In accordance with the optimized cluster structure, we computed the projected density of states (PDOS) curves[45]. As displayed in Supplementary Fig. 13, the copper atomic orbital components contribute the most to the total density of states in the energy range between −7.5 and 1.0 eV, showing copper electron delocalization in the frontier orbitals of Cu$_{40}$-H NC. In the experimental UV−Vis absorption spectrum, Cu$_{40}$-H NC exhibits the three major absorption peaks at 378, 429, and 510 nm, which corresponds well with the calculated data (Fig. 4a). The theoretical analysis of the electronic transitions and atomic components is further illustrated using a Kohn−Sham orbital energy diagram (Supplementary Fig. 14). All three spectral features are

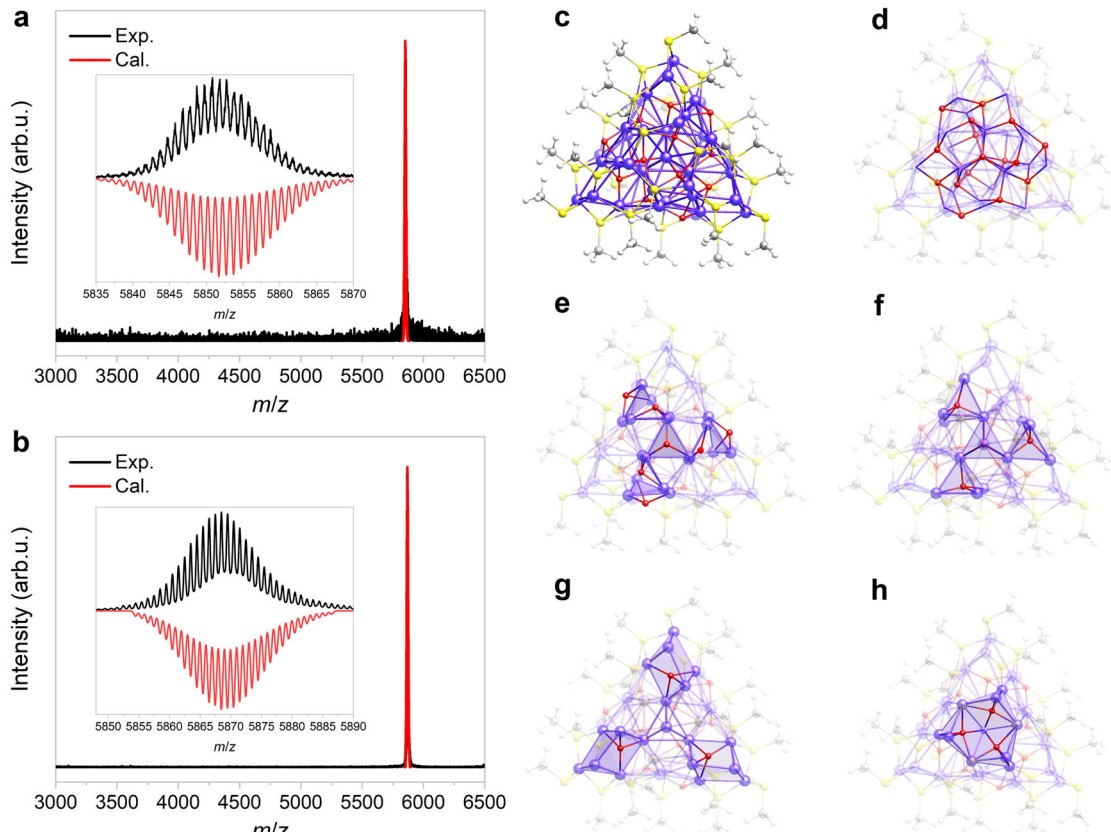

**Fig. 3 | Determination of hydrides in Cu₄₀ nanoclusters. a, b** ESI-TOF mass spectra of Cu₄₀-H NC (**a**) and Cu₄₀-D NC (**b**) in a negative mode. **c** A front view of the DFT-optimized structure for [Cu₄₀H₁₇(SCH₃)₂₄]⁻. **d** Distributions of hydride ligands in various coordination modes viewed from the back. **e** Seven μ₃-H ligands caping the triangular faces. **f** Four μ₄-H ligands in the tetrahedral units. **g** Three μ₄-H ligands in the rhombohedral units. **h** Three μ₅-H ligands in the square pyramidal units. Color labels: Cu, violet; S, yellow; C, gray; protons, white; hydrides, red. Exp., experimental data; Cal., calculated mass distributions based on the cluster's formula. Source data are provided as a Source Data file.

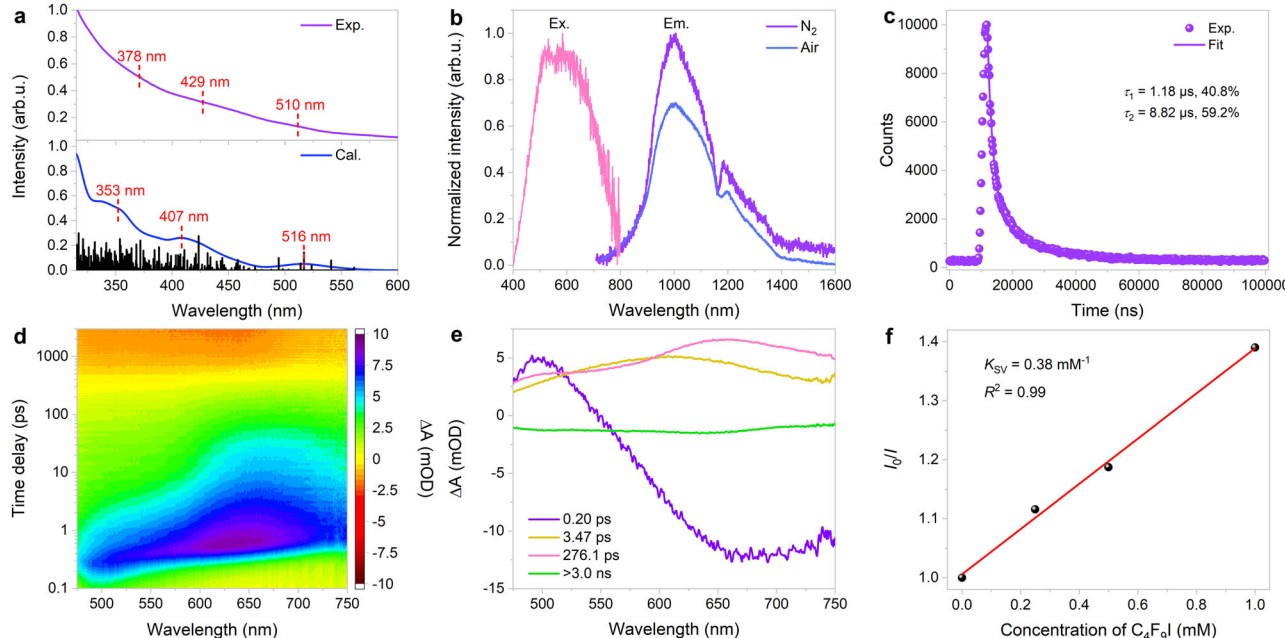

**Fig. 4 | Optical properties. a** Experimental (purple line) and calculated (blue line) UV–Vis absorption spectra of Cu₄₀-H NC. **b** Excitation (emission wavelength: 1000 nm) and emission (excitation wavelength: 600 nm) spectra of Cu₄₀-H NC in dichloromethane under N₂ or air atmosphere. **c** Emission decay (λmax = 1000 nm) of photoexcited Cu₄₀-H NC at room temperature. **d** TA data maps pumped at 400 nm in dichloromethane. **e** Species-associated spectra (from global fit analysis). **f** Stern–Volmer plot for the luminescence quenching of Cu₄₀-H NC with various concentrations of C₄F₉I. Source data are provided as a Source Data file.

primarily associated with the Cu $3d \rightarrow$ Cu $4s$ interband transitions and the ligand-to-metal charge-transfer (LMCT) transitions mixed with $d \rightarrow s/p$ metal-cluster-centered transitions[46]. The HOMO−LUMO gap is estimated to be 2.8 eV (Supplementary Fig. 15), which is comparable to previously reported data for copper nanoclusters[47]. In addition, the time-dependent UV−Vis absorption spectra suggest that $Cu_{40}$-H NC is stable in various organic solvents, including dichloromethane, acetonitrile, and N,N-dimethylacetamide (Supplementary Figs. 16–18).

Encouragingly, the copper nanocluster exhibits NIR-II emission[35] in a $N_2$-purged dichloromethane solution, with two distinct peaks at 1000 and 1174 nm, respectively (Fig. 4b). The significant decrease in emission observed upon exposing $Cu_{40}$-H NC to air is highly indicative of triplet-state photoexcitations[48]. As shown in Fig. 4c and Supplementary Fig. 19, the double-exponential fitting on the photoluminescence decay signals of $Cu_{40}$-H NC collected by time-correlated single photon counting techniques reveals two lifetime components of 1.18 μs (40.8%) and 8.82 μs (59.2%). To further understand the excited-state dynamics of $Cu_{40}$-H NC in different solvents, femtosecond transient absorption (fs-TA) experiments were performed upon excitation with 400-nm laser pulse (Fig. 4d and Supplementary Fig. 20). Based on the sequential model, the global analysis yields evolution-associated difference spectra for four major components with lifetimes ranging from 0.20 ps to > 3.0 ns, which can be categorized as three excited-state absorption (ESA) signals and one ground-state bleaching (GSB) signal (Fig. 4f). Within the first 0.6 ps, the primary ESA peak redshifts from 500 to 600 nm, suggesting the generation of the first singlet excited state from a higher excited state via internal conversion[35]. The following intersystem crossing process begins at about 3 ps and involves a redshift of the ESA signal from 600 to 650 nm, resulting in the formation of an triplet LMCT state[48]. Subsequently, an additional process with a lifetime of 276.1 ps occurs, which could be attributed to the structural relaxation of $Cu_{40}$-H NC's metal core[35]. Lastly, the GSB signal around 650 nm endures until the end of the time-delay window (3 ns), which indicates the emergence of a long-lived species (Supplementary Fig. 21). The corresponding decay process gives rise to the phosphorescence emission. fs-TA measurements in other solvents show similar decay pathways after excitation (Supplementary Figs. 22 and 23). The rapid structural relaxation of $Cu_{40}$-H NC in toluene, as compared to dichloromethane and N,N-dimethylacetamide, can be readily explained by the fact that nonpolar solvents exert a weaker stabilizing effect on the triplet LMCT state of the nanocluster[49].

To determine whether photoexcited $Cu_{40}$-H NC is capable of reducing fluoroalkyl iodide electrophiles, an important class of chemical reagents used to produce valuable fluorine-containing molecules for agricultural and pharmaceutical industries[26], we first measured the cyclic voltammogram of $Cu_{40}$-H NC under inert atmosphere (Supplementary Fig. 25). Based on the higher-energy shoulder (around 1077 nm) in its emission spectrum at an excitation wavelength of 456 nm, the photoexcited redox potential of $Cu_{40}$-H NC is estimated to be −1.73 V (vs. $Fc^+$/Fc), suggesting feasible electron transfer processes with $C_4F_9I$ ($E_{p/2}$ = −1.65 V vs. $Fc^+$/Fc, see Supplementary Fig. 26). We have also established that $C_4F_9I$ efficiently quenches the luminescence of $Cu_{40}$-H NC (Supplementary Fig. 27); the Stern−Volmer plot provides a quenching constant ($K_{SV}$) of 0.38 $mM^{-1}$ (Fig. 4f). The quenching experiments strongly support the activation of perfluoroalkyl iodides by the copper nanocluster at its photoexcited states. Importantly, the powder X-ray diffraction patterns of $Cu_{40}$-H NC remain unchanged under ambient conditions for more than two months (Supplementary Fig. 28).

## Photocatalytic performances

On account of its favorable optical properties, matched photoexcited redox potential, and high stability, $Cu_{40}$-H NC is considered as a potential photocatalyst for promoting visible-light-driven organic transformations. Since perfluoroalkyl groups can significantly increase the lipophilicity, bioavailability, and metabolic stability of bioactive compounds[28], it is highly desirable to develop efficient visible-light photocatalysis based on copper nanoclusters for the cyanofluoroalkylation of various alkenes with readily available fluoroalkyl iodides[50–52]. After examining a variety of reaction parameters, we identified a procedure that enables the desired three-component cyanofluoroalkylation (Table 1, entry 1; 78% yield). Control experiments demonstrate the importance of $Cu_{40}$-H NC, base, water, and light in the coupling reaction (entries 2–5). Notably, no product (1) was obtained at an elevated temperature in the absence of light (entry 6). Irradiation with blue-LED lamps at wavelengths of 427–467 nm results in high yields of 1 (entries 7 and 8), whereas no reaction occurs when exposed to 520-nm green-LED light (entry 9). Using cationic or neutral copper nanoclusters[53,54] instead of anionic $Cu_{40}$-H NC has a direct negative impact on cluster-based photocatalysis (entries 10 and 11). The replacement of $Cu_{40}$-H NC with a variety of Cu(I) and Cu(II) sources leads to a dramatic decrease in product yields, showing the superior photocatalytic activity of $Cu_{40}$-H NC in the visible-light-mediated cyanofluoroalkylation (entries 12–18). The catalytic efficiency of the cluster precursors, including those with ligands or phosphonium salts added, is significantly lower than that of $Cu_{40}$-H NC (entries 12–15). When the catalyst loading was reduced to 0.15 mol%, the reaction furnished the desired product in a satisfactory yield, with a turnover number (TON) of 440 (entry 19). Despite high reaction efficiency in acetonitrile (entry 20), other organic solvents, such as 1,2-dichloroethane, dimethylsulfoxide, and toluene, gave much lower yields (entry 21). Several other alkyl amines can also serve as base additives in this photocatalysis (entries 22–24), but they are not as effective as DIPEA. The addition of a non-reducing inorganic base, such as $K_3PO_4$, promotes the radical coupling (entry 25), indicating that the direct single-electron reduction of $C_4F_9I$ by photoexcited $Cu_{40}$-H NC is operative. Increasing the amount of water from two to four equivalents lowers the yield by less than 10% (entry 26), which suggests that the coupling is highly compatible with water. Similar to many photoinduced copper(I)-catalyzed radical reactions[23,55,56], the coupling enabled by $Cu_{40}$-H NC cannot proceed under an atmosphere of air (entry 27).

Under the optimized reaction conditions, a broad range of alkenes can be double-functionalized in high isolated yields (Fig. 5). In addition to 1- and 2-vinylnaphthalenes, a wide range of styrenes containing an electron-donating or -withdrawing functional group at the para-, meta-, or ortho-position of the aromatic ring perform well in the cyanofluoroalkylation, with TONs of up to 320 (Fig. 5, products 1–22). Despite the fact that a similar transformation mediated by copper(I) complexes occurred under violet light irradiation[51], the olefin substrates are mostly restricted to electron-rich styrenes, with trace amounts of products resulting from strongly electron-deficient styrenes (Fig. 5, products 7 and 18). Importantly, the cluster-based visible-light photocatalysis is tolerant of many heterocycles including pyridines, indoles, benzofurans, and benzothiophenes (Fig. 5, products 23–26). It is noteworthy that the scope of alkene substrates is not limited to styrenes; a variety of nonconjugated alkenes are also suitable reaction partners (Fig. 5, products 27–31). Benzoyl ester and aniline derivatives as well as sterically hindered aliphatic alkenes afford the desired products in synthetically useful yields.

With regard to the electrophile scope, a number of linear and branched perfluoroalkyl iodides, including trifluoromethyl iodide, are found to be highly reactive (Fig. 5, products 32–36). Under the standard reaction conditions, α,α-difluoro carbonyl compounds can be successfully prepared without the use of any external oxidants (Fig. 5, products 37–41), demonstrating the practicality of this cluster-based approach in organic synthesis.

To further illustrate the robustness of $Cu_{40}$-H NC in visible-light photocatalysis, we chose an α-bromo isobutyric ester as an electrophile partner in the three-component radical coupling, resulting in a

**Table 1 | Effects of reaction parameters on the copper-catalyzed cyanofluoroalkylation induced by visible light**

2-vinylnaphthalene + $C_4F_9I$ (3.0 equiv.) + TMSCN (3.0 equiv.) → Cu$_{40}$-H NC (0.3 mol%), hν (blue LED, 456 nm), DIPEA, H$_2$O, DMA, N$_2$, r.t., 20 h "standard conditions" → product **1** (CN, $C_4F_9$)

| Entry | Change from "standard conditions" | Yield (%)[a] | TON[b] |
|---|---|---|---|
| 1 | none | 78 | 260 |
| 2 | no catalyst | <5 | – |
| 3 | no base | <5 | – |
| 4 | no water | 11 | 37 |
| 5 | no light | <5 | – |
| 6 | no light, 80 °C | <5 | – |
| 7 | hν (427 nm) | 75 | 250 |
| 8 | hν (467 nm) | 70 | 233 |
| 9 | hν (520 nm) | <5 | – |
| 10 | Cu$_8$ NC (0.3 mol%), instead of Cu$_{40}$-H NC | 13 | 43 |
| 11 | Cu$_{54}$ NC (0.3 mol%), instead of Cu$_{40}$-H NC | 31 | 103 |
| 12 | [Cu(CH$_3$CN)$_4$]BF$_4$ (5.0 mol%), instead of Cu$_{40}$-H NC | 11 | 2.2 |
| 13[c] | [Cu(CH$_3$CN)$_4$]BF$_4$ (5.0 mol%), instead of Cu$_{40}$-H NC | 23 | 4.6 |
| 14[d] | [Cu(CH$_3$CN)$_4$]BF$_4$ (5.0 mol%), instead of Cu$_{40}$-H NC | 11 | 2.2 |
| 15[e] | [Cu(CH$_3$CN)$_4$]BF$_4$ (5.0 mol%), instead of Cu$_{40}$-H NC | 13 | 2.6 |
| 16 | CuCl (5.0 mol%), instead of Cu$_{40}$-H NC | 15 | 3.0 |
| 17 | Cu(OAc)$_2$ (5.0 mol%), instead of Cu$_{40}$-H NC | 15 | 3.0 |
| 18 | Cu(OTf)$_2$ (5.0 mol%), instead of Cu$_{40}$-H NC | 12 | 2.4 |
| 19 | Cu$_{40}$-H NC (0.15 mol%) | 66 | 440 |
| 20 | CH$_3$CN, instead of DMA | 71 | 237 |
| 21 | DCE, DMSO, or toluene as solvent | <40 | <133 |
| 22 | NEt$_3$, instead of DIPEA | 70 | 233 |
| 23 | NHEt$_2$, instead of DIPEA | 65 | 217 |
| 24 | NBnMe$_2$, instead of DIPEA | 46 | 153 |
| 25 | K$_3$PO$_4$, instead of DIPEA | 39 | 130 |
| 26 | H$_2$O (4.0 equiv.) | 69 | 230 |
| 27 | under air, instead of nitrogen | <5 | – |

Standard conditions: 2-vinylnaphthalene (0.10 mmol, 1.0 equiv.), C$_4$F$_9$I (3.0 equiv.), TMSCN (3.0 equiv.), Cu$_{40}$-H NC (0.3 mol%), DIPEA (4.0 equiv.), and H$_2$O (2.0 equiv.) in anhydrous DMA (1.0 mL) under nitrogen atmosphere at room temperature with blue-LED light irradiation (456 nm) for 20 h. [a]Yield was determined by [1]H NMR of the crude product using 1,2-dibromoethane as an internal standard. [b]TONs were calculated based on the crude NMR yield of **1**. TMSCN, trimethylsilyl cyanide; DIPEA, *N*,*N*-diisopropylethylamine; DMA, *N*,*N*-dimethylacetamide; DCE, 1,2-dichloroethane; DMSO, dimethylsulfoxide; Bn, benzyl; Cu$_8$ NC, [Cu$_8$H(9*H*-carbazole-9-carbodithioate)$_6$](PF$_6$)[53]; Cu$_{54}$ NC, [Cu$_{54}$S$_{13}$O$_6$(tBuS)$_{20}$(tBuSO$_3$)$_{12}$][54]. [c]2,4-DMBTH (6.0 mol%) was added. [d]PPh$_3$ (6.0 mol%) was added. [e]PPh$_4$Br (6.0 mol%) was added.

satisfactory yield of cyanated product **42** (Fig. 6a). This class of bromide electrophiles were underdeveloped in previous copper-based photocatalytic systems[50,51]. Furthermore, no two-component bromoalkylation of alkenes occurred in the absence of a catalyst. Given that only a small amount of iodofluoroalkylated products were observed upon substituting C$_4$F$_9$I for the bromide electrophile (Table S4)[57], we concluded that the formation of an electron donor–acceptor complex[58,59] may not be the primary mechanism by which electrophiles are activated to produce alkyl radicals in this cluster-based photocatalysis. A radical-clock experiment with a cyclopropyl-substituted alkene clearly implies a radical addition into the double bond prior to the copper-mediated cyanation process (Fig. 6b).

Based on the ESI-TOF mass data of the reaction mixture (Supplementary Fig. 29) and the UV–Vis absorption spectrum of the recycled catalyst (Supplementary Fig. 30), it can be determined that Cu$_{40}$-H NC remains largely intact throughout the photocatalytic process. Given that both DIPEA and water are critical to the cyanoalkylation, we propose a catalytic cycle involving the activation of TMSCN under the

basic conditions (Fig. 6c). Upon exposure to blue-LED light, photoexcited Cu$_{40}$-H NC undergoes single-electron transfer with an alkyl halide to produce a mixed-valence cluster intermediate, a reactive alkyl radical (R'•), and an iodide anion. The subsequent trapping of R'• by a terminal alkene generates a secondary alkyl radical (R''•) for further copper-mediated functionalization processes. In the presence of DIPEA and water, the neutral mixed-valence cluster intermediate reacts with TMSCN to yield a anionic cyanide-ligated cluster intermediate and trimethylsilanol with a strong Si–O single bond[30]. The radical recombination between R''• and the cyanide-ligated cluster intermediate affords the C–C coupling product and regenerates the original copper(I) nanocluster[60–62]. The high photostability of Cu$_{40}$-H NC in the presence of DIPEA excludes the possibility of the amine base acting as a sacrificial reagent to reduce the photoexcited copper nanocluster (Supplementary Fig. 31).

We present here the synthesis and characterization of an atomically precise Cu$_{40}$ nanocluster in a $C_3$ symmetry, which is the first example of anionic copper nanoclusters with NIR-II emission. Due to its wide optical absorption range and excellent

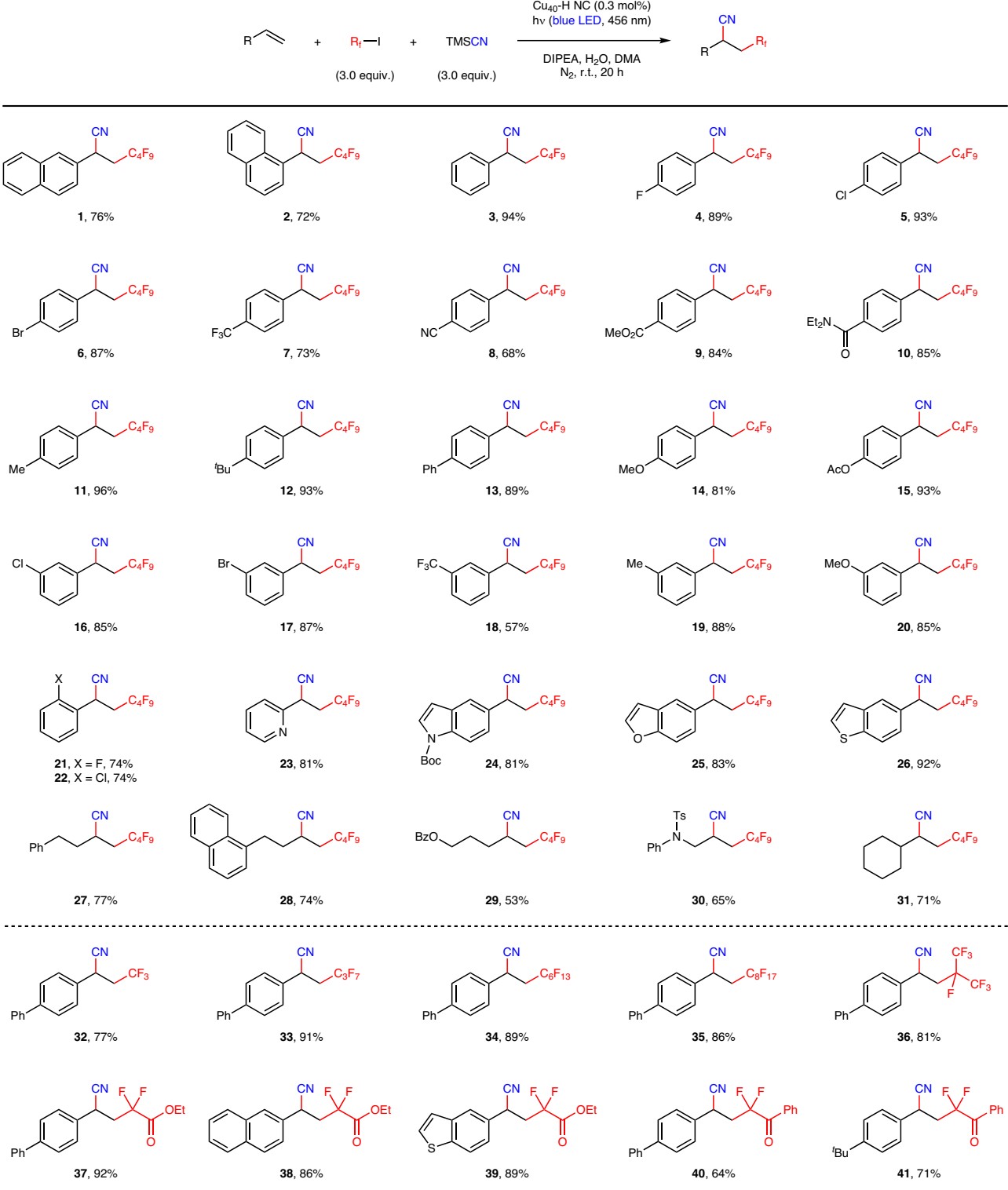

**Fig. 5 | Substrate scope for cyanofluoroalkylation of alkenes.** Reaction conditions: alkene (0.10 mmol, 1.0 equiv.), fluoroalkyl iodide (3.0 equiv.), TMSCN (3.0 equiv.), $Cu_{40}$-H NC (0.3 mol%), DIPEA (4.0 equiv.), and $H_2O$ (2.0 equiv.) in anhydrous DMA (1.0 mL) under nitrogen atmosphere at room temperature with blue-LED light irradiation (456 nm) for 20 h. For each entry number (in bold), data are reported as isolated yields. TMSCN, trimethylsilyl cyanide; DIPEA, *N,N*-diisopropylethylamine; DMA, *N,N*-dimethylacetamide; *t*Bu, *tert*-butyl; Ac, acetyl; Boc, *tert*-butoxycarbonyl; Bz, benzoyl; Ts, *p*-toluenesulfonyl.

reducing ability in its photoexcited states, this stable 2,4-DMBT-protected copper(I) cluster can be employed as a highly effective photoredox catalyst for the cyanofluoroalkylation of alkenes. Beyond providing the first cluster-catalyzed three-component radical coupling, we have discovered that the current method has a fairly broad substrate scope under blue-LED irradiation conditions. Further efforts are being made to investigate other types of visible-light-driven radical reactions with cheap and scalable nanocluster catalysts for sustainable and practical organic synthesis.

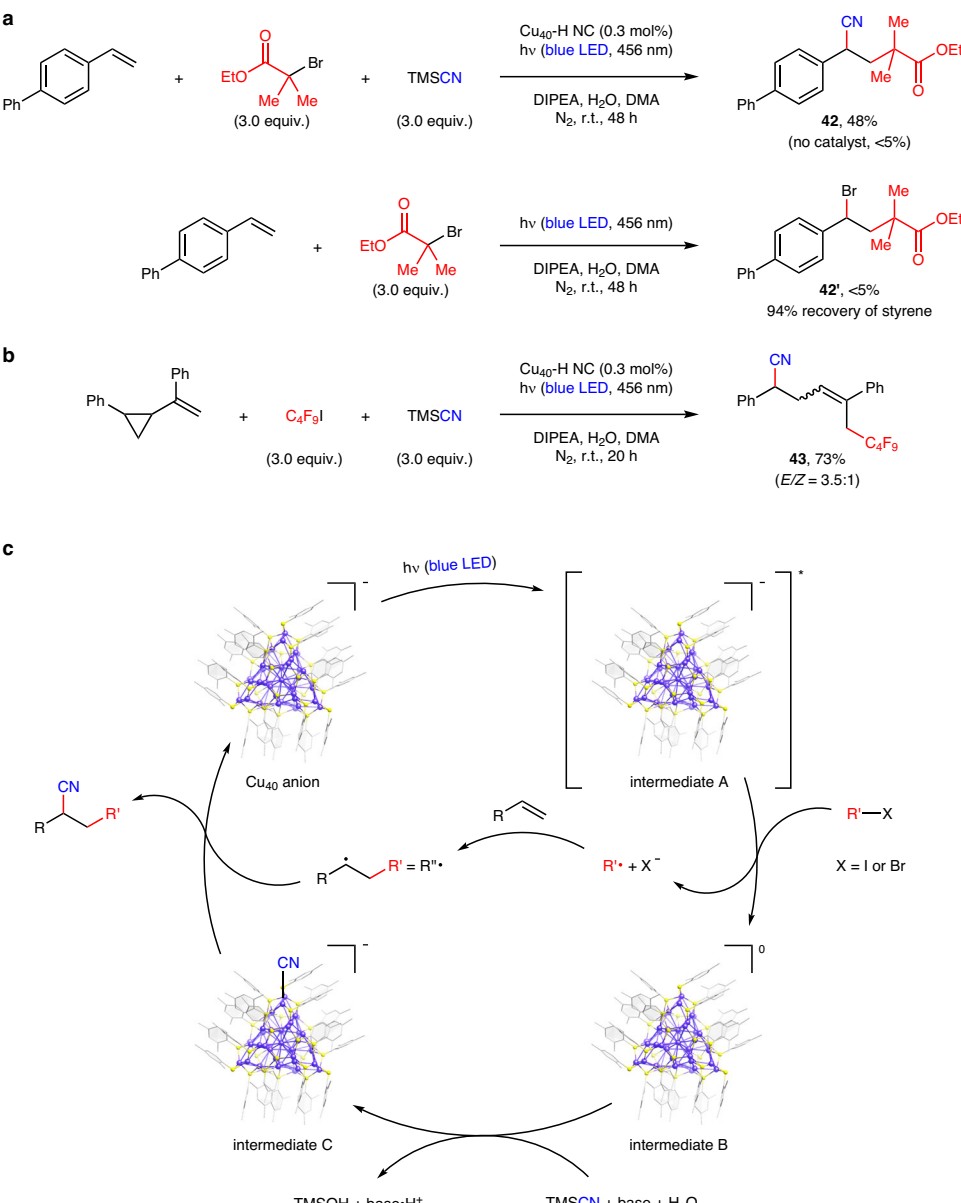

**Fig. 6 | Mechanistic studies. a** Visible-light-induced radical coupling with an alkyl bromide electrophile. **b** A radical-clock experiment. **c** Plausible mechanism for copper nanocluster-based photocatalysis.

## Methods

### General considerations

Unless otherwise noted, materials were purchased from commercial suppliers and used as received. Tetrakis(acetonitrile)copper(I) hexafluorophosphate ([Cu(CH$_3$CN)$_4$]PF$_6$, 98%) and tetraphenylphosphonium bromide (PPh$_4$Br, 98%) were purchased from Leyan Co., Ltd. Sodium borohydride (NaBH$_4$, 99.8% purity) and sodium borohydride-d$_4$ (NaBD$_4$, 98% purity) were purchased from Sigma-Aldrich. 2,4-dimethylbenzenethiol (2,4-DMBTH, 98% purity) was purchased from Saan Chemical Technology Co., Ltd. Methanol (>99% purity), dichloromethane (>99% purity), toluene (>99% purity), hexanes (>99% purity), and acetonitrile (>99% purity) were purchased from Tianjin Kemiou Chemical Reagent Co., Ltd.

X-ray diffraction data of the crystal was collected using synchrotron radiation ($\lambda$ = 0.67043 Å) on beamline 17B1 at the National Facility for Protein Science Shanghai in the Shanghai Synchrotron Radiation Facility, Shanghai, People's Republic of China. The diffraction data reduction and integration were performed by the HKL3000 software.

Powder X-ray diffraction patterns were recorded on a Rigaku Ultima IV X-ray diffractometer (CuK$\alpha$, $\lambda$ = 1.5418 Å), operating at 40 kV and 30 mA. The measurement parameters included a scan speed of 10° min$^{-1}$, a step size of 0.05°, and a scan range of 2$\theta$ from 3° to 50°.

Scanning electron microscope images and energy dispersive spectroscopy mapping were collected on an EM-30 AX PLUS microscope (South Korea, COXEM company).

ESI-TOF mass spectrometry data were recorded on a Waters Q-TOF mass spectrometer using a Z-spray source. High-resolution ESI mass measurements were performed on a Bruker impact II high-resolution LC-QTOF mass spectrometer.

All UV−Vis absorption spectra were acquired in the 200–800 nm range using a Cary 3500 spectrophotometer (Agilent). Steady-state emission spectra were obtained on an Edinburgh FLS1000 spectrophotometer.

$^1$H NMR spectra were recorded on a Bruker 500 (500 MHz) or Bruker 400 (400 MHz) spectrometer in chloroform-$d$. Chemical shifts were quoted in parts per million (ppm) referenced to 0.0 ppm of

tetramethyl silane. $^{13}C$ NMR spectra were recorded on a Bruker 500 or Bruker 400 spectrometer in chloroform-$d$ with complete proton decoupling. $^{19}F$ and $^{31}P$ NMR spectra were recorded on a Bruker 500 or Bruker 400 spectrometer.

Cyclic voltammograms were performed on a CHI760E electrochemistry workstation. Regular 3-electrode systems were used. Measurements were recorded in an acetonitrile solution of $Bu_4NClO_4$ (0.1 M) at a scan rate of 100 mV s$^{-1}$ under the protection of $N_2$ using a glassy carbon disk (d = 0.3 cm) as a working electrode and a platinum plate (1 cm × 1 cm) as a counter electrode. An Ag/AgCl (3 M KCl) electrode was used as a reference electrode in all the experiments, and its potential (0.46 V vs. Fc$^+$/Fc) was calibrated with the ferrocenium/ferrocene (Fc$^+$/Fc) redox couple.

### Synthesis of $Cu_{40}H_{17}(2,4\text{-DMBT})_{24}](PPh_4)$ nanocluster

[$Cu(CH_3CN)_4$]$PF_6$ (50 mg, 0.13 mmol) and PPh$_4$Br (20 mg, 0.048 mmol) were dissolved in acetonitrile (5 mL). Then, 2,4-DMBTH (10 μL, 0.074 mmol) was introduced to the reaction. After stirring for 10 min, freshly prepared NaBH$_4$ (50 mg, 1.3 mmol) in an ice-cold methanol solution (5 mL) was added instantaneously. The solvent was evaporated after the reduction for 5 h, the remaining solid was dissolved in dichloromethane and filtered. Red block-like crystals of $Cu_{40}$-H NC suitable for single-crystal X-ray analysis were obtained by slow vapor diffusion of hexanes into 5-mL dichloromethane-toluene (1:1 v/v) of the nanoclusters at −4 °C for one week.

### Gram-scale synthesis

[$Cu(CH_3CN)_4$]$PF_6$ (4.65 g, 12.5 mmol) and PPh$_4$Br (1.89 g, 4.51 mmol) were dissolved in acetonitrile (450 mL). Then, 2,4-DMBTH (0.930 mL, 6.90 mmol) was introduced to the reaction. After stirring for 10 min, freshly prepared NaBH$_4$ (4.70 g, 124 mmol) in an ice-cold methanol solution (450 mL) was added instantaneously. The solvent was evaporated after the reduction for 5 h, the remaining solid was dissolved in dichloromethane and filtered. Red block-like crystals of $Cu_{40}$-H NC suitable for single-crystal X-ray analysis were obtained by slow vapor diffusion of hexanes into 400-mL dichloromethane-toluene (1:1 v/v) of the nanoclusters at −4 °C for one week.

### Synthesis of $Cu_{40}D_{17}(2,4\text{-DMBT})_{24}](PPh_4)$ nanocluster

[$Cu(CH_3CN)_4$]$PF_6$ (50 mg, 0.13 mmol) and PPh$_4$Br (20 mg, 0.048 mmol) were dissolved in acetonitrile (5 mL). Then, 2,4-DMBTH (10 μL, 0.074 mmol) was introduced to the reaction. After stirring for 10 min, freshly prepared NaBD$_4$ (55 mg, 1.3 mmol) in an ice-cold methanol solution (5 mL) was added instantaneously. The solvent was evaporated after the reduction for 5 h, the remaining solid was dissolved in dichloromethane and filtered. Red block-like crystals of $Cu_{40}$-D NC were obtained by slow vapor diffusion of hexanes into 5-mL dichloromethane-toluene (1:1 v/v) of the nanoclusters at −4 °C for one week.

### Computational studies

For the experimental complex, [$Cu_{40}H_{17}(SR)_{24}$]$^-$ (R = 2,4-Me$_2C_6H_3$), the aryl groups were replaced by the −CH$_3$ in theoretical computations to reduce the computational cost without affecting the interfacial bond strength.

Calculations of UV−Vis absorption spectra and Kohn−Sham (K−S) molecular orbitals: Gaussian 16 package[63] was used to obtain the optimized geometry by Perdew-Burke-Ernzerhof hybrid functional (PBE0)[64] method with Grimme's BJ-damped variant of DFT-D3 empirical dispersion[65,66]. The pseudopotential basis set LANL2DZ and all-electron def2-SVP were used for Cu atoms and other atoms (H, C, and S), respectively. The time dependent density functional theory method implemented in Gaussian 16 was used to compute the simulated spectra using the same functional, empirical dispersion and basis sets as above. K−S orbital analysis was performed for identifying the

atomic orbital contribution to each molecular orbital using Multiwfn 3.8 program[67,68].

Based on the hydrides assigned by X-ray diffraction data, all possible positions of the hydrides in [$Cu_{40}H_{17}(SR)_{24}$]$^-$ (R = CH$_3$) were predicted. The energetically favored structure was then fully optimized without any constraints, and the resulting structure is the predicted final model structure shown in Fig. 3c.

### TA measurements

Femtosecond transient absorption measurements were performed at room temperature using a Spectra Physics Tsunami Ti:Sapphire (Coherent; 800 nm, 150 fs, 7 mJ pulse$^{-1}$, and 1 kHz repetition rate) as the laser source and a Helios spectrometer (Ultrafast Systems LLC). Briefly, the 800-nm output pulse from the regenerative amplifier was split in two parts. 95% of the output from the amplifier is used to pump a TOPAS optical parametric amplifier, which generated a wavelength-tunable laser pulse from 400 to 850 nm as pump beam in a Helios transient absorption setup (Ultrafast Systems Inc.). The 400-nm pump beam was used for the measurements. The remaining 5% of the amplified output was focused onto a sapphire crystal to generate a white-light continuum used for probe beam in our measurements (420 to 850 nm). The pump beam was depolarized and chopped at 1 kHz, and both pump and probe beams were overlapped in the sample for magic angle transient measurements. Samples were vigorously stirred in all the measurements. Species-associated spectra were obtained by fitting the principal kinetics deduced from single value decomposition analysis.

### Stern−Volmer experiments

An Edinburgh FLS1000 spectrophotometer was used for luminescence quenching experiments. Linear regression of $I_0/I$ against concentration was performed in Origin. All samples for the luminescence test were prepared in the glovebox, and the measurements were performed at room temperature. $Cu_{40}$-H NC solution (0.025 mM) in acetonitrile was excited at 456 nm and the emission was collected at 1077 nm. For each quenching experiment, a certain volume of the stock solution was added into a 4-mL solution of $Cu_{40}$-H NC (0.025 mM) in a 10-mm quartz cuvette with a screw cap.

### Procedure for cluster-based photocatalysis

To a 10-mL flame-dried Schlenk tube under N$_2$ atmosphere were added $Cu_{40}$-H NC (2.0 mg, 0.3 mol%), alkene (0.10 mmol, 1.0 equiv.), fluoroalkyl iodide (0.30 mmol, 3.0 equiv.), TMSCN (29.8 mg, 0.30 mmol, 3.0 equiv.), DIPEA (69.7 μL, 0.40 mmol, 4.0 equiv.), H$_2$O (3.6 μL, 0.20 mmol, 2.0 equiv.), and anhydrous DMA (1.0 mL) sequentially. The reaction mixture was irradiated by 40-watt Kessil PR160L-456 blue-LED lamps at room temperature for 20 h. After irradiation, the reaction mixture was diluted with saturated brine (10 mL) and extracted four times with ethyl acetate (4 × 5 mL). The combined organic layers were dried over anhydrous Na$_2SO_4$ and concentrated under vacuum. The residue was purified by flash column chromatography on silica gel to afford the corresponding product.

### Reporting summary

Further information on research design is available in the Nature Portfolio Reporting Summary linked to this article.

## Data availability

The data supporting the finding of this study are available in this article and the Supplementary Information. Crystallographic data for $Cu_{40}$-H NC have been deposited at the Cambridge Crystallographic Data Centre, under deposition number CCDC 2263793. Copies of the data can be obtained free of charge via https://www.ccdc.cam.ac.uk/structures/. Source data are provided in this paper. Source data are provided with this paper.

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

## Acknowledgements

The authors gratefully acknowledge The University of Hong Kong, the Research Grants Council of the Hong Kong Special Adminis-trative Region, People's Republic of China (RGC: 27301820, J.H., 17313922, J.H.), the Croucher Foundation, the Innovation and Technology Commission (HKSAR, China), the National Natural Science Foundation of China (No. 22201236, J.H.) for their financial support. We also thank S.-Q. Zang, X.-J. Zhai, and the staff of BL17B1 beamline at the National Facility for Protein Science of the Shanghai Synchrotron Radiation Facility for assistance during data collection. V.W.-W.Y. acknowledges support from the Key Program of the Major Research Plan on "Architectures, Functionalities and Evolution of Hierarchical Clusters" of the National Natural Science Foundation of China (No. 91961202, V.W.-W.Y.). Publication made possible in part by support from the HKU Libraries Open Access Author Fund sponsored by the HKU Libraries.

## Author contributions

L.-J.L. synthesized the copper nanoclusters and conducted most of the experiments for characterization, with assistance from Y.L. M.-M.Z. optimized the photocatalytic reaction conditions. Z.D. and D.L.P. col-lected and analyzed the data from transient absorption measurements, and L.-L.Y. performed SCXRD experiments. V.W.-W.Y. provided insights into cluster characterization and mechanistic studies. J.H. directed the project and wrote the manuscript with the contributions from L.-J.L. and M.-M.Z.

## Competing interests

The authors declare no competing interests.
