## [Peer Review File · Nature Communications]

NIR-II emissive anionic copper nanoclusters with intrinsic photoredox activity in single-electron transferREVIEWER COMMENTS

Reviewer #1 (Remarks to the Author):

Comments on the manuscript (NCOMMS-23-52611)

Recommendation:

The authors successfully synthesized atomically precise NIR-II emissive anionic Cu₄₀NC and explored their potential in photocatalytic applications involving the three-component couplings of olefins, fluoroalkyl iodides, and trimethyl cyanide. The nanocluster-based photocatalytic process demonstrated notable compatibility with a broad spectrum of alkenes and fluoroalkyl iodides. The supporting information within the manuscript is comprehensive, presenting all required data.

Despite the successful preparation of the NIR-II emissive anionic Cu₄₀NC structure, the catalysis reactions were found to be less impressive. Existing literature suggests that the combination of activated alkyl iodides and aliphatic amine bases (e.g., Et₃N or DIPEA) in polar aprotic solvents can easily form Electron Donor-Acceptor (EDA) complexes upon visible-light irradiation, generating the corresponding alkyl radicals (Ref: Eur. J. Org. Chem. 2018, 6391, and related reports).

In contrast, the manuscript posits that anionic Cu₄₀NC serves as a potential photocatalyst, facilitating single electron transfer with fluoroalkyl iodides to produce alkyl radicals. This claim raises questions regarding the absence of EDA complex formation and challenges the understanding of how anionic Cu₄₀NC functions as a genuine photocatalyst.

In order to provide a better understanding the use of unactivated alkyl halides (X=Br or Cl) could be beneficial: not only for shedding light into the reaction mechanism but also to address a more challenging research question.

Interestingly, while the Cu₄₀NC cluster exhibits NIR-II emissive behavior, the photocatalytic reaction (alkene, RfI & TMSCN) is conducted under the blue-LEDs region. The apparent discrepancy prompts inquiry into the correlation between NIR-II emissive properties and the catalytic applications of the cluster.

In summary, the structural aspects and the preparation/characterization of the cluster are of interest. However, the catalytic applications and the mechanistic understanding, as presented, may fall short of meeting the standards set by Nature Communications.

Further clarification on the mechanistic aspects, particularly addressing the omission of EDA complex formation and the choice of electrophiles, is recommended for a comprehensive and compelling manuscript.

Cluster part:

- In the literature, the electron count of Ag₄₀ clusters is characterized as a 2, 14, or 16-electron super-atomic system. However, the [Cu₄₀H₁₇(2,4-DMBT)₂₄]-nanocluster exhibits an unusual negative electron count. The authors are encouraged to elucidate this discrepancy or reassess the number of hydride ions. Typically, the electron counts for copper nanoclusters are reported as zero or positive. Additionally, in the full Electrospray Ionization Mass Spectrometry (ESI MS) spectrum, the theoretical spectrum hinders the experimental spectrum. To clarify the results, the theoretical spectrum should be omitted or distinctly separated from the experimental data."

Catalytic part:

- Previously, different copper catalysts have been reported in these three-component reactions (even asymmetric ones). The authors need to point out that what are the key differences and advantages if compared to the previous copper systems (see, related reports ACS Catal. 2019, 9, 4470, Chem. Commun., 2017,53, 12317-12320).
- Table 1, in the absence of a catalyst, a trace amount (<5%) of the desired product is obtained. Is in this case, the electron-donor-acceptor (EDA-complex) involving a two-component coupling reaction observed? It is known that fluoroalkyl iodides and DIPEA bases can form EDA complexes to form the products under blue-light irradiation.

- In addition, Cu₄₀ exhibited better catalytic activity than simple Cu-salts (entries 10-12, Table 1). Simple Cu-salts (e.g., CuCl or Cu(OAc)₂) with coordinated ligands, and counter-anions of the Cu could have significant effects on the reactivity of the reactions. Therefore, it would be more rational to test the catalytic activity of the copper complex, which forms an intermediate during the preparation of Cu₄₀ cluster structure. For example, the authors should examine other Cu-complexes, such as, a) [Cu(CH₃CN)₄]₂·BF₄, (b) copper complexes resulting from the reaction of [Cu(CH₃CN)₄]₂·BF₄ with 2,4-dimethylbenzenethiol, (c) copper complexes resulting from the reaction of [Cu(CH₃CN)₄]₂·BF₄ with PPh₃, and (d) copper complexes resulting from the reaction of [Cu(CH₃CN)₄]₂·BF₄ with tetraphenylphosphonium bromide (PPh₄Br), respectively.
- It's essential to rationalize the role of water in the three-component couplings. Have the authors tried an inorganic base instead of an amine base?
- Although various photophysical studies have been conducted and described in the supporting information, performing the radical clock experiment to further support the radical pathway is recommended.
- The proposed mechanism scheme should be presented in the main text instead of SI

Reviewer #2 (Remarks to the Author):

Li-Juan Liu et al. demonstrated the synthesis of an anionic copper nanocluster with NIR-II emission that can be employed in the photocatalytic applications of radical coupling processes. This new nanocluster exhibits exceptional activity in single-electron transfer upon photoexcitation, leading to the development of blue-light-induced cyanofluoroalkylation of alkenes with a broad substrate scope. The authors thoroughly analyzed the structure of the Cu₄₀ nanocluster and exploited its photochemical properties using transient absorption spectroscopy. The successful establishment of cluster-based photocatalysis in organic synthesis is essential for the advancement of the nanocluster fields as well as for probing the comprehensive mechanistic investigation of transition metal-catalyzed organic reactions from a new perspective. Furthermore, developing a rare example of three-component radical coupling reactions under photoirradiation with a scalable and recyclable copper nanocluster meets the criteria for sustainable chemical synthesis. The findings in the current work hold significant implications for establishing interdisciplinary research projects that integrate organic chemistry and nanoscience. I would be very enthusiastic about publication for a prestigious journal like Nature Communications, after the authors address the following minor issues:

1. To further characterize the composition of Cu₄₀-H, EDS mapping and ³¹P NMR analysis are highly recommended.
2. To support the significance of using anionic copper nanoclusters in photocatalysis, the control experiment with a neutral or cationic copper nanocluster is very useful.
3. The previous synthetic method, which used traditional copper(I) complexes as photocatalysts, was incompatible with electron-deficient styrenes (see ACS Catal. 2019, 9, 4470). It is necessary to investigate the reactivity of 3-(trifluoromethyl)styrene in the current cluster-based photocatalysis.
4. The authors chose 2,4-dimethylbenzenethiolate as a protecting ligand to prepare copper nanoclusters. Can it be replaced by other types of thiolate ligands? The authors should provide more information in the synthesis section, as the steric and electronic effects of protecting ligands have huge influence on the cluster synthesis and their chemical properties.
5. I think some closely related work about Cu nanoclusters should be noted such as 10.1021/acs.accounts.5b00375; Nature Communications 14 (2023) 6413; Angew. Chem. Int. Ed., 62, (2023), e202306822; Angew. Chem. Int. Ed., 62, (2023), e202306849

Jian He, Ph.D.
Assistant Professor
Email: jianhe@hku.hk
Phone: (852) 3910 2193
Fax: (852) 2857 1586

March 6, 2024

We sincerely appreciate the reviewers' time and efforts in helping us improve the quality of our work. We conducted additional experiments and revised the manuscript in response to their constructive feedback and suggestions.

Reviewer #1

(1) The authors successfully synthesized atomically precise NIR-II emissive anionic Cu₄₀NC and explored their potential in photocatalytic applications involving the three-component couplings of olefins, fluoroalkyl iodides, and trimethyl cyanide. The nanocluster-based photocatalytic process demonstrated notable compatibility with a broad spectrum of alkenes and fluoroalkyl iodides. The supporting information within the manuscript is comprehensive, presenting all required data.

Response: We appreciate the reviewer's positive comments about our work.

(2) Despite the successful preparation of the NIR-II emissive anionic Cu₄₀NC structure, the catalysis reactions were found to be less impressive. Existing literature suggests that the combination of activated alkyl iodides and aliphatic amine bases (e.g., Et₃N or DIPEA) in polar aprotic solvents can easily form Electron Donor-Acceptor (EDA) complexes upon visible-light irradiation, generating the corresponding alkyl radicals (Ref: *Eur. J. Org. Chem.* **2018**, 6391, and related reports).

In contrast, the manuscript posits that anionic Cu₄₀NC serves as a potential photocatalyst, facilitating single electron transfer with fluoroalkyl iodides to produce alkyl radicals. This claim raises questions regarding the absence of EDA complex formation and challenges the understanding of how anionic Cu₄₀NC functions as a genuine photocatalyst.

In order to provide a better understanding the use of unactivated alkyl halides (X=Br or Cl) could be beneficial: not only for shedding light into the reaction mechanism but also to address a more challenging research question.

Response: We thank the reviewer for the very important comments. We agree that exploring other alkyl halide electrophiles to rule out an EDA-complex pathway is critical to determining whether Cu₄₀-H is a genuine photocatalyst. Therefore, we carried out a photocatalytic reaction with a bromide electrophile and a control experiment to support radical formation via single-electron transfer with the nanocluster photocatalyst. We updated the manuscript as follows:

“To further illustrate the robustness of **Cu₄₀-H** in visible-light photocatalysis, we chose an α -bromo isobutyric ester as an electrophile partner in the three-component radical coupling, resulting in a satisfactory yield of cyanated product **42** (Fig. 6a). This class of bromide electrophiles were underdeveloped in previous copper-based photocatalytic systems^{50,51}. Furthermore, no two-component bromoalkylation of alkenes occurred in the absence of a catalyst. Given that only a small amount of iodofluoroalkylated products were observed upon substituting C₄F₉I for the bromide electrophile (Table S4)⁵⁷, we concluded that the formation of an electron donor–acceptor complex^{58,59} is not the primary mechanism by which electrophiles are activated to produce alkyl radicals in this cluster-based photocatalysis.”

(3) Interestingly, while the Cu₄₀NC cluster exhibits NIR-II emissive behavior, the photocatalytic reaction (alkene, R_fI & TMSCN) is conducted under the blue-LEDs region. The apparent discrepancy prompts inquiry into the correlation between NIR-II emissive properties and the catalytic applications of the cluster.

Response: We thank the reviewer for the important concern. We employed blue-LED irradiation to photoexcite the Cu₄₀ nanocluster on the basis of the major responses in the UV–Vis absorption spectrum (Fig. 4a). The NIR-II emission data was primarily used to estimate the photoexcited redox potential of the nanocluster and conduct quenching studies with an iodide electrophile.

“Based on the higher-energy shoulder (around 1077 nm) in its emission spectrum at an excitation wavelength of 456 nm, the photoexcited redox potential of **Cu₄₀-H** is estimated to be -1.73 V (vs. Fc⁺/Fc), suggesting feasible electron transfer processes with C₄F₉I ($E_{p/2} = -1.65$ V vs. Fc⁺/Fc, see Supplementary Fig. 26). We have also established that C₄F₉I efficiently quenches the luminescence of **Cu₄₀-H** (Supplementary Fig. 27); the Stern–Volmer plot provides a quenching constant (K_{SV}) of 0.38 mM⁻¹ (Fig. 4f). The quenching experiments strongly support the activation of perfluoroalkyl iodides by the copper nanocluster at its photoexcited states.”

(4) In summary, the structural aspects and the preparation/characterization of the cluster are of interest. However, the catalytic applications and the mechanistic understanding, as presented, may fall short of meeting the standards set by Nature Communications.

Further clarification on the mechanistic aspects, particularly addressing the omission of EDA complex formation and the choice of electrophiles, is recommended for a comprehensive and compelling manuscript.

Response: We thank the reviewer for the very constructive comments. The revised manuscript includes additional investigations on bromide electrophiles, as well as control experiments to assess the possibility of EDA complex formation. Please refer to the following paragraphs:

“To further illustrate the robustness of **Cu₄₀-H** in visible-light photocatalysis, we chose an α -bromo isobutyric ester as an electrophile partner in the three-component radical coupling, resulting in a satisfactory yield of cyanated product **42** (Fig. 6a). This class of bromide electrophiles were underdeveloped in previous copper-based photocatalytic systems^{50,51}. Furthermore, no two-component bromoalkylation of alkenes occurred in the absence of a catalyst. Given that only a small amount of iodofluoroalkylated products were observed upon substituting C₄F₉I for the bromide electrophile (Table S4)⁵⁷, we concluded that the formation of an electron donor–acceptor complex^{58,59} is not the primary mechanism by which electrophiles are activated to produce alkyl radicals in this cluster-based photocatalysis.”

Table S4. Investigation of visible-light-induced two-component coupling via the formation of an electron donor–acceptor complex

Entry	Catalyst	Yield of 1 (%)	Yield of 1' (%)
1	Cu₄₀-H	78	<5
2	no catalyst	<5	9

Standard conditions: 2-vinylnaphthalene (0.1 mmol, 1.0 equiv.), C₄F₉I (3.0 equiv.), TMSCN (3.0 equiv.), **Cu₄₀-H** (0.3 mol%) or without catalyst, DIPEA (4.0 equiv.), and H₂O (2.0 equiv.) in anhydrous DMA (1.0 mL) under nitrogen atmosphere at room temperature with blue-LED light irradiation (456 nm) for 20 h. Yield was determined by ¹H NMR of the crude product using 1,2-dibromoethane as an internal standard.

(5) In the literature, the electron count of Ag₄₀ clusters is characterized as a 2, 14, or 16-electron super-atomic system. However, the [Cu₄₀H₁₇(2,4-DMBT)₂₄]-nanocluster exhibits an unusual negative electron count. The authors are encouraged to elucidate this discrepancy or reassess the number of hydride ions. Typically, the electron counts for copper nanoclusters are reported as zero or positive. Additionally, in the full Electrospray Ionization Mass Spectrometry (ESI MS) spectrum, the theoretical spectrum hinders the experimental spectrum. To clarify the results, the theoretical spectrum should be omitted or distinctly separated from the experimental data.

Response: We thank the reviewer for the important concerns. In general, superatomic nanoclusters consist of metal centers with a valence state of zero. Using [Ag₄₀(2,4-DMBT)₂₄(PPh₃)₈H₁₂]²⁺ as one example (*J. Am. Chem. Soc.* **2019**, *141*, 11905), this Ag₄₀ nanocluster has two Ag(0) atoms and is characterized as a 2-electron superatomic system. In another case, [Cu₃₁(4-MeO-PhC≡C)₂₁(dppe)₃](ClO₄)₂ with four Cu(0) atoms belongs to a 4-electron superatomic system (*J. Am. Chem. Soc.* **2023**, *145*, 10355). Cu(0)-containing nanoclusters are difficult to prepare because copper has a substantially lower M(I)/M(0) reduction potential (0.52 V) than silver (0.80 V), making it sensitive to air during the synthetic process. As a result, the vast majority of copper nanoclusters have a valence state of +1. **Cu₄₀-H** is a monovalent copper(I) nanocluster with a negative charge; similar situations can be found in other copper nanoclusters.

Entry	Molecular formula	Properties	Ref.
1	[Cu ₄₀ H ₁₇ (2,4-DMBT) ₂₄] ⁻	NIR-II emissive	Our work
2	[Cu ₂₅ H ₁₀ (SC ₆ H ₃ Cl ₂) ₁₈] ³⁻	non-emissive	ACS Nano 2019 , 13 , 5975
3	[Cu ₄₁ H ₂₅ (SC ₆ H ₃ F ₂) ₁₅ Cl ₃ (P(C ₆ H ₄ F) ₃) ₆] ²⁻	non-emissive	Adv. Sci. 2023 , 2307085
4	[Cu ₃₂ H ₈ (PET) ₂₄ Cl ₂] ²⁻	non-emissive	JACS 2020 , 142 , 13974

We updated the ESI-TOF mass spectra to separate the theoretical and experimental data:

Fig. 3 | Determination of hydrides in Cu_{40} nanoclusters. a,b ESI-TOF mass spectra of $\text{Cu}_{40}\text{-H}$ (a) and $\text{Cu}_{40}\text{-D}$ (b) in a negative mode.

Supplementary Figure 10. ESI-TOF mass spectra of $\text{Cu}_{40}\text{-H}$ in a positive mode.

(6) Previously, different copper catalysts have been reported in these three-component reactions (even asymmetric ones). The authors need to point out that what are the key differences and advantages if compared to the previous copper systems (see, related reports *ACS Catal.* **2019**, *9*, 4470, *Chem. Commun.*, **2017**, *53*, 12317-12320).

Response: We appreciate the very helpful comment from the reviewer. In the previous *ACS Catalysis* paper, trifluoromethyl substituted styrenes are incompatible with the photoinduced, copper-catalyzed three-component coupling reactions. To demonstrate the broad substrate scope in cluster-based photocatalysis, we included the cyanofluoroalkylation of 3-(trifluoromethyl)styrene in Fig. 5 and provided a thorough discussion as follows:

“In addition to 1- and 2-vinylnaphthalenes, a wide range of styrenes containing an electron-donating or -withdrawing functional group at the *para*-, *meta*-, or *ortho*-position of the aromatic ring perform well in the cyanofluoroalkylation, with TONs of up to 320 (Fig. 5, products **1–22**). Despite the fact that a similar transformation mediated by copper(I) complexes occurred under violet light irradiation⁵¹, the olefin substrates are mostly restricted to electron-rich styrenes, with trace amounts of products resulting from strongly electron-deficient styrenes (Fig. 5, products **7** and **18**).”

We also examined an alkyl bromide electrophile which was not reported in the previous copper systems (see, *Chem. Commun.* **2017**, *53*, 12317; *ACS Catal.* **2019**, *9*, 4470) and added the following discussion to the revised manuscript:

“To further illustrate the robustness of **Cu₄₀-H** in visible-light photocatalysis, we chose an α -bromo isobutyric ester as an electrophile partner in the three-component radical coupling, resulting in a satisfactory yield of cyanated product **42** (Fig. 6a). This class of bromide electrophiles were underdeveloped in previous copper-based photocatalytic systems^{50,51}.”

(7) Table 1, in the absence of a catalyst, a trace amount (<5%) of the desired product is obtained. Is in this case, the electron-donor-acceptor (EDA-complex) involving a two-component coupling reaction observed? It is known that fluoroalkyl iodides and DIPEA bases can form EDA complexes to form the products under blue-light irradiation.

Response: We appreciate the very helpful comment from the reviewer. The two-component iodofluoroalkylation via an EDA complex was observed in the absence of **Cu₄₀-H**, but the product yield was quite low (Table S4). In the literature (see, *Org. Chem. Front.* **2023**, *10*, 3861), the same transformation was achieved through the use of a frustrated radical pair ([PhN⁺Me₂][B(C₆F₅)₃⁻]). The major pathway to produce fluoroalkyl radicals is halogen-atom transfer. No product was detected without B(C₆F₅)₃ (see Table 1, entry 4 in the *Org. Chem. Front.* paper). In general, the efficiency of radical formation via an EDA complex can be enhanced by increasing the irradiation power and reaction temperature (see, *Angew. Chem. Int. Ed.* **2018**, *57*, 814). Under our mild reaction conditions, we believe that the formation of EDA complexes is not the predominant mechanism for generating fluoroalkyl radicals.

Table S4. Investigation of visible-light-induced two-component coupling via the formation of an electron donor–acceptor complex

Entry	Catalyst	Yield of 1 (%)	Yield of 1' (%)
1	Cu₄₀-H	78	<5
2	no catalyst	<5	9

Standard conditions: 2-vinylnaphthalene (0.1 mmol, 1.0 equiv.), C₄F₉I (3.0 equiv.), TMSCN (3.0 equiv.), **Cu₄₀-H** (0.3 mol%) or without catalyst, DIPEA (4.0 equiv.), and H₂O (2.0 equiv.) in anhydrous DMA (1.0 mL) under nitrogen atmosphere at room temperature with blue-LED light irradiation (456 nm) for 20 h. Yield was determined by ¹H NMR of the crude product using 1,2-dibromoethane as an internal standard.

(8) In addition, Cu₄₀ exhibited better catalytic activity than simple Cu-salts (entries 10-12, Table 1). Simple Cu-salts (e.g., CuCl or Cu(OAc)₂) with coordinated ligands, and counter-anions of the Cu could have significant effects on the reactivity of the reactions. Therefore, it would be more rational to test the catalytic activity of the copper complex, which forms an intermediate during the preparation of Cu₄₀ cluster structure. For example, the authors should examine other Cu-complexes, such as, a) [Cu(CH₃CN)₄] \cdot BF₄, (b) copper complexes resulting from the reaction of [Cu(CH₃CN)₄] \cdot BF₄ with 2,4-dimethylbenzenethiol, (c) copper complexes resulting from the reaction of [Cu(CH₃CN)₄] \cdot BF₄ with PPh₃, and (d) copper complexes resulting from the reaction of [Cu(CH₃CN)₄] \cdot BF₄ with tetraphenylphosphonium bromide (PPh₄Br), respectively.

Response: We appreciate the helpful suggestion from the reviewer. We agree that both ligands and counter-anions could have significant effects on the photocatalytic activity of copper species. Therefore, we performed the control experiments as suggested by the reviewer and included the following discussion in the revised manuscript:

“The replacement of **Cu₄₀-H** with a variety of Cu(I) and Cu(II) sources leads to a dramatic decrease in product yields, showing the superior photocatalytic activity of **Cu₄₀-H** in the visible-light-mediated cyanofluoroalkylation (entries 12–18). The catalytic efficiency of the cluster precursors, including those with ligands or phosphonium salts added, is significantly lower than that of **Cu₄₀-H** (entries 12–15).”

Entry	Change from “standard conditions”	Yield ^a (%)	TON ^b
12	[Cu(CH ₃ CN) ₄] \cdot BF ₄ (5.0 mol%), instead of Cu₄₀-H	11	2.2
13 ^c	[Cu(CH ₃ CN) ₄] \cdot BF ₄ (5.0 mol%), instead of Cu₄₀-H	23	4.6
14 ^d	[Cu(CH ₃ CN) ₄] \cdot BF ₄ (5.0 mol%), instead of Cu₄₀-H	11	2.2
15 ^e	[Cu(CH ₃ CN) ₄] \cdot BF ₄ (5.0 mol%), instead of Cu₄₀-H	13	2.6
16	CuCl (5.0 mol%), instead of Cu₄₀-H	15	3.0
17	Cu(OAc) ₂ (5.0 mol%), instead of Cu₄₀-H	15	3.0
18	Cu(OTf) ₂ (5.0 mol%), instead of Cu₄₀-H	12	2.4

Standard conditions: 2-vinylnaphthalene (0.1 mmol, 1.0 equiv.), C₄F₉I (3.0 equiv.), TMSCN (3.0 equiv.), **Cu₄₀-H** (0.3 mol%), DIPEA (4.0 equiv.), and H₂O (2.0 equiv.) in anhydrous DMA (1.0 mL) under nitrogen atmosphere at room temperature with blue-LED light irradiation (456 nm) for 20 h. ^aYield was determined by ¹H NMR of the crude product using 1,2-dibromoethane as an

internal standard. ^bTONs were calculated based on the crude NMR yield of **1**. TMSCN, trimethylsilyl cyanide; DIPEA, *N,N*-diisopropylethylamine; DMA, *N,N*-dimethylacetamide; DCE, 1,2-dichloroethane; DMSO, dimethylsulfoxide; Bn, benzyl; **Cu₈**, [Cu₈H(9*H*-carbazole-9-carbodithioate)₆](PF₆)⁵³; **Cu₅₄**, [Cu₅₄S₁₃O₆(*t*BuS)₂₀(*t*BuSO₃)₁₂]⁵⁴. ^c2,4-DMBTH (6.0 mol%) was added. ^dPPh₃ (6.0 mol%) was added. ^ePPh₄Br (6.0 mol%) was added.

(9) It's essential to rationalize the role of water role in the three-component couplings. Have the authors have tried an inorganic base instead of an amine base?

Response: We thank the review for the important concern. We updated the manuscript to include an inorganic base in Table 1:

“Several other alkyl amines can also serve as base additives in this photocatalysis (entries 22–24), but they are not as effective as DIPEA. The addition of a non-reducing inorganic base, such as K₃PO₄, promotes the radical coupling (entry 25), indicating that the direct single-electron reduction of C₄F₉I by photoexcited **Cu₄₀-H** is operative.”

Entry	Change from “standard conditions”	Yield ^a (%)	TON ^b
22	NEt ₃ , instead of DIPEA	70	233
23	NHET ₂ , instead of DIPEA	65	217
24	NBnMe ₂ , instead of DIPEA	46	153
25	K₃PO₄, instead of DIPEA	39	130

Due to the relatively low solubility of K₃PO₄ in DMA, water remains a major promoter in this cluster-based photocatalysis. Without water, the yield of **1** decreases considerably. Given that there was no product in the absence of water under standard conditions, it is most likely that OH⁻ and PO₄³⁻ activate TMSCN to produce cyanide anions in situ.

(10) Although various photophysical studies have been conducted and described in the supporting information, performing the radical clock experiment to further support the radical pathway is recommended.

Response: We thank the reviewer for the constructive suggestion. Consequently, the radical-clock experiment was conducted, and the following discussion was incorporated into the updated manuscript:

“A radical-clock experiment with a cyclopropyl-substituted alkene clearly implies a radical addition into the double bond prior to the copper-mediated cyanation process (Fig. 6b).”

(11) The proposed mechanism scheme should be presented in the main text instead of SI.

Response: We thank the reviewer for the helpful suggestion. The proposed catalytic cycle was included in Fig. 6c.

Reviewer #2

(1) The findings in the current work hold significant implications for establishing interdisciplinary research projects that integrate organic chemistry and nanoscience. I would be very enthusiastic about publication for a prestigious journal like Nature Communications, after the authors address the following minor issues:

Response: We thank the reviewer for the very positive comments on our work.

(2) To further characterize the composition of Cu₄₀-H, EDS mapping and ³¹P NMR analysis are highly recommended.

Response: We thank the reviewer for the constructive comments. Accordingly, we carried out additional experiments to further analyze the composition of Cu₄₀-H and validate the existence of tetraphenylphosphonium cation in the nanocluster:

“After one week of slow vapor diffusion of hexanes into a dichloromethane-toluene solution, dark red crystals of Cu₄₀-H suitable for single-crystal X-ray diffraction (SCXRD) measurements were obtained on a gram scale in a high yield of 60% (based on Cu) (Supplementary Figs. 1 and 2). Elemental mapping shows that Cu, S, and P elements are uniformly distributed throughout its block-like crystals (Supplementary Fig. 3).”

“The counterion (PPh_4^+) was also successfully identified by ESI-TOF mass spectrometry in a positive mode and ^{31}P NMR studies (Supplementary Figs. 10 and 11).”

Supplementary Figure 3. SEM image and elemental mapping of **Cu₄₀-H**. EDS mapping images of Cu, S, and P are represented in magenta, cyan, and yellow, respectively. Scale bar, 50 μm.

Supplementary Figure 11. ^{31}P NMR spectrum of **Cu₄₀-H** in CDCl_3 .

(3) To support the significance of using anionic copper nanoclusters in photocatalysis, the control experiment with a neutral or cationic copper nanocluster is very useful.

Response: We thank the reviewer for the important suggestion. We carried out the control experiments to compare the photocatalytic activity of cationic and neutral copper nanoclusters in the three-component coupling and included a discussion in the manuscript:

“Using cationic or neutral copper nanoclusters^{53,54} instead of anionic **Cu₄₀-H** has a direct negative impact on cluster-based photocatalysis (entries 10 and 11). The replacement of **Cu₄₀-H** with a variety of Cu(I) and Cu(II) sources leads to a dramatic decrease in product yields, showing the superior photocatalytic activity of **Cu₄₀-H** in the visible-light-mediated cyanofluoroalkylation (entries 12–18).”

Entry	Change from “standard conditions”	Yield ^a (%)	TON ^b
1	none	78	260
2	no catalyst	<5	–
3	no base	<5	–
4	no water	11	37
5	no light	<5	–
6	no light, 80 °C	<5	–
7	hv (427 nm)	75	250
8	hv (467 nm)	70	233
9	hv (520 nm)	<5	–
10	Cu₈ (0.3 mol%), instead of Cu₄₀-H	13	43
11	Cu₅₄ (0.3 mol%), instead of Cu₄₀-H	31	103

Standard conditions: 2-vinylnaphthalene (0.1 mmol, 1.0 equiv.), C₄F₉I (3.0 equiv.), TMSCN (3.0 equiv.), **Cu₄₀-H** (0.3 mol%), DIPEA (4.0 equiv.), and H₂O (2.0 equiv.) in anhydrous DMA (1.0 mL) under nitrogen atmosphere at room temperature with blue-LED light irradiation (456 nm) for 20 h. ^aYield was determined by ¹H NMR of the crude product using 1,2-dibromoethane as an internal standard. ^bTONs were calculated based on the crude NMR yield of **1**. TMSCN,

trimethylsilyl cyanide; DIPEA, *N,N*-diisopropylethylamine; DMA, *N,N*-dimethylacetamide; DCE, 1,2-dichloroethane; DMSO, dimethylsulfoxide; Bn, benzyl; **Cu₈**, [Cu₈H(9*H*-carbazole-9-carbodithioate)₆](PF₆)⁵³; **Cu₅₄**, [Cu₅₄S₁₃O₆(^{*t*}BuS)₂₀(^{*t*}BuSO₃)₁₂]⁵⁴.

(4) The previous synthetic method, which used traditional copper(I) complexes as photocatalysts, was incompatible with electron-deficient styrenes (see ACS Catal. 2019, 9, 4470). It is necessary to investigate the reactivity of 3-(trifluoromethyl)styrene in the current cluster-based photocatalysis.

Response: We thank the reviewer for the very important suggestion. As a result, 3-(trifluoromethyl)styrene was employed to survey the substrate scope of the newly developed nanocluster-based photocatalysis (see updated Fig. 5, product **18**). We included a detailed analysis in the manuscript:

“In addition to 1- and 2-vinylnaphthalenes, a wide range of styrenes containing an electron-donating or -withdrawing functional group at the *para*-, *meta*-, or *ortho*-position of the aromatic ring perform well in the cyanofluoroalkylation, with TONs of up to 320 (Fig. 5, products **1–22**). Despite the fact that a similar transformation mediated by copper(I) complexes occurred under violet light irradiation⁵¹, the olefin substrates are mostly restricted to electron-rich styrenes, with trace amounts of products resulting from strongly electron-deficient styrenes (Fig. 5, products **7** and **18**).”

(5) The authors chose 2,4-dimethylbenzenethiolate as a protecting ligand to prepare copper nanoclusters. Can it be replaced by other types of thiolate ligands? The authors should provide more information in the synthesis section, as the steric and electronic effects of protecting ligands have huge influence on the cluster synthesis and their chemical properties.

Response: We thank the reviewer for the important question and comments. 2,4-Dimethylbenzenethiolate cannot be replaced by other thiolate ligands in the current cluster synthesis. Therefore, we included a detailed discussion in the manuscript:

“The air- and moisture-stable Cu₄₀ nanocluster was synthesized by directly reducing [Cu(CH₃CN)₄]BF₄ with NaBH₄ in the presence of 2,4-dimethylbenzenethiol (2,4-DMBTH) and tetraphenylphosphonium bromide (PPh₄Br) under ambient conditions. **Other less sterically hindered benzenethiols and aliphatic thiols, such as 4-methylbenzenethiol and 2-phenylethanethiol, failed to produce Cu₄₀ nanoclusters, highlighting the significance of weak interactions from 2,4-DMBT in the nanocluster formation.**”

(6) I think some closely related work about Cu nanoclusters should be noted such as 10.1021/acs.accounts.5b00375; Nature Communications 14 (2023) 6413; Angew. Chem. Int. Ed., 62, (2023), e202306822; Angew. Chem. Int. Ed., 62, (2023), e202306849.

Response: We thank the reviewer for the important suggestion. The first reference has already been cited in the manuscript (ref 44). The remaining three references were included in the current revision.

5. Wu, Q.-J. *et al.* Atomically precise copper nanoclusters for highly efficient electroreduction of CO₂ towards hydrocarbons via breaking the coordination symmetry of Cu site. *Angew. Chem. Int. Ed.* **62**, e202306822 (2023).
9. Zhang, C. *et al.* Dynamic and transformable Cu₁₂ cluster-based C–H⋯π-stacked porous supramolecular frameworks. *Nat. Commun.* **14**, 6413 (2023).
43. Luo, G.-G. *et al.* Total structure, electronic structure and catalytic hydrogenation activity of metal-deficient chiral polyhydride Cu₅₇ nanoclusters. *Angew. Chem. Int. Ed.* **62**, e202306849 (2023).
44. Dhayal, R. S., van Zyl, W. E. & Liu, C. W. Polyhydrido copper clusters: synthetic advances, structural diversity, and nanocluster-to-nanoparticle conversion. *Acc. Chem. Res.* **49**, 86–95 (2016).

REVIEWERS' COMMENTS

Reviewer #2 (Remarks to the Author):

The authors have revised the article thoroughly, and all the queries were addressed. The additional experiments provided are also convincing. It can be accepted now.

*Department of Chemistry and State Key Laboratory of Synthetic Chemistry
Room 103, Hui Oi Chow Science Building, Pokfulam Road, Hong Kong*

Jian He, Ph.D.
Assistant Professor
Email: jianhe@hku.hk
Phone: (852) 3910 2193
Fax: (852) 2857 1586

April 27, 2024

We sincerely appreciate the reviewers' time and efforts in helping us improve the quality of our work.

Reviewer #2

The authors have revised the article thoroughly, and all the queries were addressed. The additional experiments provided are also convincing. It can be accepted now.

Response: We thank the reviewer for his/her very positive comments on our manuscript.